# Working from Home during the COVID-19 Pandemic and Its Effects on Diet, Sedentary Lifestyle, and Stress

**DOI:** 10.3390/nu14194006

**Published:** 2022-09-27

**Authors:** Merve Güney Coşkun, Rabia İclal Öztürk, Ayşegül Yabacı Tak, Nevin Sanlier

**Affiliations:** 1Department of Nutrition and Dietetics, Faculty of Health Sciences, Istanbul Medipol University, Istanbul 34810, Turkey; 2Department of Nutrition and Dietetics, Institute of Health Sciences, Istanbul Medipol University, Istanbul 34810, Turkey; 3Research Institute for Health Sciences and Technologies (SABITA), Istanbul Medipol University, Istanbul 34810, Turkey; 4Department of Biostatistics and Medical Informatics, Faculty of Medicine, Bezmialem Vakıf University, Istanbul 34093, Turkey; 5Department of Nutrition and Dietetics, School of Health Sciences, Ankara Medipol University, Ankara 06570, Turkey

**Keywords:** COVID-19, dietary behavior, sedentary lifestyle, stress, working from home

## Abstract

Many companies switched to working from home (WFH) after the COVID-19 pandemic. This paper aimed to examine the changes in dietary behavior, body weight, sedentary lifestyle, and stress in individuals who practice WFH. A cross-sectional, web-based questionnaire was administered between March and May 2021 and included socio-demographic characteristics, anthropometric measurements, WFH arrangement, changes in diet, sedentary lifestyle, and stress status. A total of 328 individuals (260 women, 68 men), aged 31.3 ± 8.3 years with a BMI of 24.9 ± 4.6 kg/m^2^, participated in the study. The questionnaire revealed that the daily working time increased with WFH. The majority of the individuals (59.1%) gained weight. The average daily sedentary time and the Perceived Stress Scale score increased significantly. The daily sedentary time and Non-Healthy Diet Index scores were higher in individuals who gained weight (*p* < 0.05). A multinominal regression model revealed that increased body weight was less likely in individuals with underweight and normal BMI classifications. Normal BMI, stable work shifts, and no physical activity were positive predictors for gaining weight. These results suggest that WFH may have significant negative effects on physical and mental status of individuals.

## 1. Introduction

The coronavirus disease 2019 (COVID-19) caused by severe acute respiratory syndrome coronavirus 2 (SARS-CoV-2) was first reported in late December 2019 and rapidly spread all over the world [1]. To curb the pandemic, beside many other precautions, people were asked to avoid crowded places, not travel during rush hours, stay at home, and engaging in working from home (WFH). Almost 4 out of 10 workers in Europe started working from home [2]. Many daily personal routines changed after the COVID-19 pandemic, and the concept of a “new normal” emerged in many areas. Companies have rapidly adapted to this “new normal” concept approach positively, with WFH as a “new normal way of working” not only during the COVID-19 pandemic period but also beyond [3]. However, employees have negative as well as positive experiences regarding WFH [4]. Gender, number of people and children at home, and position cause differential impacts on people’s lives who WFH [4]. Additionally, the transition to a WFH routine per se puts physical and mental burdens on individuals [5]. In particular, the necessity of staying at home for a long time during the pandemic has been associated with general stress, leading to changes in physical activity and eating habits [6,7].

According to the results of a study on occupational health of individuals who started WFH, the most common physical health problem was weight gain, with a rate of 41% [8]. Other occupational health problems associated with the intensity of WFH have been reported, such as musculoskeletal pain, problems arising from isolation and a closed environment, depression, work fatigue, and burnout [8]. On the other hand, the people involved in the study stated that they prefer WFH because it allows for a wider variety of daily activities, less stress in the work environment, flexible working hours, more time to spend with the family, and it being easier to maintain a better and healthier lifestyle [8].

To date, many studies conducted during lockdowns have found unhealthy lifestyle changes in individuals’ physical activity status and eating habits [9,10,11]. The COVID-19 pandemic has been directly associated with lifestyle changes in adults such as weight gain, increased consumption of unhealthy foods, uncontrolled eating, frequent snacking between meals, unhealthy dietary habits, less exercise, and more than 6 h of sedentary time [9,10,11,12]. In contrast, a Spanish study has shown that quarantine helped adults adopt healthier diets [12]. A recent study found that 66% of their respondents reported no change in their healthy eating compared to before the lockdown period [13]. Conversely, in another study, 45% of the participants stated that their eating habits changed positively, 32% negatively, and 23% neither positively nor negatively [14]. Additionally, decreased physical activity and increased sedentary time are other results of lockdown orders and stay-at-home calls [15]. Negative psychological effects have been shown to be another consequence of quarantine [16,17]. Research on mental health has revealed that COVID-19 causes numerous emotional consequences, including stress, depression, irritability, insomnia, fear, confusion, anger, frustration, boredom, and stigma [16].

The consequences of COVID-19 restrictions may be interrelated and interconnected, and it is recommended that they be studied together [13,18]. Although there are many studies investigating the behavioral changes prominent in this period in the general population, it is known that additional studies are needed on individuals in specific groups and conditions. Given the growing evidence that WFH is unlikely to be a temporary phenomenon, there is a need to determine changes in lifestyle [19,20].

We hypothesized that WFH would influence dietary behavior, body weight, sedentary lifestyle, and stress. Therefore, we aimed to explore potential relationships between WFH and lifestyle changes reflected in dietary behavior, physical activity, and stress during the COVID-19 pandemic.

## 2. Materials and Methods

### 2.1. Population and Study Design

The study adopted the snowball sampling technique, and data collection was carried out with an anonymous online self-administrated questionnaire between March 2021 and May 2021. Following the construction process of the questionnaire form, it was uploaded to Google Forms and delivered to the volunteered participants to whom the researchers reached through individual relationships via social media channels (i.e., Facebook, Twitter, Instagram, and LinkedIn), email, and messaging applications with a special target of employees practicing WFH.

Participants between the ages of 18 and 65 who started WFH during the COVID-19 pandemic were included in the study. A total of 328 participants completed all mandatory questions. The Google form questionnaire only allowed one response per user, eliminating multiplication of individual records.

### 2.2. Ethical Standards Disclosure

This study was conducted according to the guidelines laid down in the Declaration of Helsinki, and all procedures involving human subjects were approved by the Non-Interventional Clinical Research Ethics Committee of Istanbul Medipol University (E-10840098-772.02-6563). Written informed consent was obtained from all subjects, and they were asked for permission to use and publish the data from the study before starting the online questionnaire.

### 2.3. Measures

The questionnaire had six parts;
(1)Socio-demographic (sex, age, education level, occupation, married status);(2)Anthropometric characteristics (4 questions);(3)Working from home arrangements (8 questions);(4)Changes in nutritional habits and instruments (53 questions);(5)Physical activity and sedentary lifestyle changes (6 questions);(6)Stress status before and after working from home (21 questions).


#### 2.3.1. Anthropometric Measurements

Since the participants could not be interviewed face-to-face due to the COVID-19 pandemic, the body mass index (BMI) was calculated on the basis of their statements. The reported body weight (kg) was divided by the square of their reported height (m) (BMI = kg/m^2^). According to the World Health Organization (WHO) cut-offs, BMI was divided into four groups (underweight, normal, overweight, and obese) [21]. Additionally, the weight changes of the participants were recorded by asking whether they gained or lost weight and how much it was.

#### 2.3.2. Working from Home Arrangements 

WFM arrangements including participants’ certain work schedules and changes in their work shifts were questioned. The participants were asked how long they had been practicing WFH, for how many days, and regarding the changes in the start and end of working hours.

#### 2.3.3. Nutritional Habits and Instruments 

Dietary behavior changes, main meal consumption, and snacking frequency were inquired upon.

To determine changes in dietary behavior, the second part of the validated Dietary Habits and Nutrition Beliefs Questionnaire–KomPAN 2014, developed by the Behavioral Conditions of Nutrition Team, was adopted in this study [22].The description of food frequency (times/day) intakes of selected 10 healthy and 14 unhealthy food components were determined for diet indexes. Pro-Healthy Diet Index (pHDI-10) scores and Non-Healthy Diet Index (nHDI-14) scores of the participants in the study were calculated according to the procedure, which has previously been described in detail elsewhere [22]. pHDI-10 score concentrates on food with potentially beneficial effects on health, such as whole meal bread, buckwheat, oats, wholegrain pasta or other coarse-ground groats, milk, fermented milk products, fresh cheese curd products, white meat, fish, pulse-based foods, fruit, and vegetables. On the other hand, nHDI-14 score includes the food negatively impacting health such as white bread and bakery products, white rice, white pasta, fine-ground groats, fast foods, fried foods, butter, lard, fatty cheese, cured meat, smoked sausages, hot-dogs, red meat, sweets, tinned meats, sweetened carbonated or still drinks, energy drinks, and alcoholic beverages. The selection and classification of foods into “healthy” and “unhealthy” was adapted on the basis of current recommendations of the Turkish Dietary Guideline [23].

The frequency of food consumption questions (6-point Likert scale, never/1–3 times a month/once a week/few times a week/once a day/few times a day) were based on 24 food items for “before” and “during” the pandemic, separately. To standardize the way of interpreting and analyzing the results, cumulative food frequency (times/day) was used in calculations and expressed on a scale from 0 to 100 points, as recommended. The interpretation of indexes is intuitive—the higher the value of the index, the higher the intensity of beneficial or harmful characteristics for health. To evaluate the internal consistency of scores in the sample, Cronbach’s alpha was found to be 0.845 for nHDI-14 and 0.738 for pHDI-10, which was considered acceptable.

#### 2.3.4. Physical Activity and Sedentary Lifestyle Changes

Questions regarding physical activity changes were asked in accordance with adherence to WHO physical activity recommendations [24]. Participants who performed at least 150–300 min of moderate-intensity aerobic or 75–150 min of vigorous-intensity aerobic physical activity during the week were considered to be doing physical activity.

Additionally, a question about how many hours they keep seated (while using public transport; working at home; using a computer, tablet, or phone; watching television; etc.) on an average day was added to assess changes in the daily sedentary time.

#### 2.3.5. Perceived Stress Scale (PSS)

In the last part, the Perceived Stress Scale (PSS), which can be used to measure individuals’ subjective stress perceptions, was included in the survey to determine the stress level changes of individuals [25]. The degree to which individuals believe their life was unpredictable, uncontrollable, and overloaded right before the pandemic started and during the pandemic was evaluated with a 10-item scale (PSS-10) with 6 positive items and 4 negative items rated on a 5-point Likert scale. The scores ranged from 0 to 40. A high score indicates an excess of one’s perception of stress.

### 2.4. Statistical Analyses 

The analytic plan was specified and identified before, and any data-driven analyses were discussed appropriately.

Data analyses were performed using the statistical package IBM SPSS Statistics for Windows, Version 22.0 (IBM Corp.: Armonk, NY, USA, released 2011). Before statistical analyses, the normality of variable distribution was checked with a Kolmogorov– Smirnov test. Descriptive statistics were conducted. Data are expressed as frequency (*n*) and percentage (%) for categorical data and median (min–max) for continuous data.

The comparison of the independent variables that did not show normal distribution according to the groups was examined with the Mann–Whitney U test, while the Wilcoxon signed rank test was used to calculate the differences before and during the COVID-19 pandemic. The Fisher–Freeman–Halton test was used to examine the variables affecting weight changes during WFH. In addition, multinomial logistic regression analyses were applied for risk assessment of the interactions of these factors. Odds ratios (OR), 95% confidence intervals (CI) for odds ratios (lower–upper bound), and significance values (*p*-values) were given in all estimated multinomial logistic regression models. Statistical significance for all tests was set at a *p*-value of 0.05.

## 3. Results

A total of 328 participants (*n* = 260, 79.3% female and *n* = 68, 20.7% male) completed the questionnaire (Table 1). The mean age of the individuals participating in the study was 31.3 ± 8.3 years. The mean BMI was calculated as 24.9 ± 4.6 kg/m^2^, and 47.9% of the participants were classified as normal. Individuals reporting weight gain during the pandemic (*n* = 194; 59.1%) were in the majority.

Teaching was the most common occupation among participants, with a rate of 32.6%. This ranking is followed by architecture and engineering with 17.7%, and business and financial operations with 16.8%. The number of days those individuals work at home varied. However, 39.6% of the respondents reported that they work from home 5 days a week. Forty-six percent of participants reported that their daily working time increased with a switch to WFH.

The impact of WFH during the COVID-19 pandemic on BMI; pHDI-10, nHDI-14, and PSS scores; and daily sedentary time is shown in Figure 1. Mean BMI was calculated as 24.0 kg/m^2^ before the pandemic, which increased to 24.9 kg/m^2^ during WFH time (*p* < 0.001). Regarding eating habits, 34.5% stated that their diet became unhealthier, while 22.9% declared that they had started to eat healthier. However, when the pHDI-10 and nHDI-14 scores were compared before and during the COVID-19 pandemic, no significant difference was found (pHDI-10 *p* = 0.105; nHDI-14 *p* = 0.210). Considering the daily snack sessions before and after the COVID-19 pandemic, the number of those who had consumed snacks after midday and at night increased, and the number of those who had snacks before midday decreased (*p* < 0.001). Snacking increased by 10% overall (Appendix A). The PSS score increased significantly from 18.2 points to 22.9 points after the pandemic (*p* < 0.001). The average daily sedentary time increased from 7.7 h to 10.6 h (*p* < 0.001).

The average weight changes; BMI; pHDI-10, nHDI-14, and PSS scores; and daily sedentary time of individuals with body weight gain and loss during the COVID-19 pandemic were compared (Table 2). No significant difference was found in the average weight changes between groups (*p* = 0.927). While the current BMI of individuals who gained weight during the pandemic was 25.7 kg/m^2^, the current BMI of those who lost weight was 22.8 kg/m^2^ (*p* < 0.001). The nHDI-14 score of those who gained weight was calculated to be 9.2 points higher than those who lost weight (*p* < 0.001). The pHDI-10 scores of individuals who gained and lost weight did not differ (*p* = 0.452). No significant differences were found in the PSS score between the weight gain and loss groups (*p* = 0.416). Average daily sedentary time was higher in those who gained weight (*p* = 0.028). 

In a bivariate analyses (Table 3), BMI, changes in nHDI-14 score, daily sedentary time, and physical activity showed a significant correlation with the outcome variable (weight changes) (*p* < 0.001). There was a significant difference between BMI groups according to weight change (*p* < 0.001). It was concluded that most of the people who gained weight (*n* = 112; 57.7%) during the pandemic were classified as overweight and obese. Additionally, those who lost weight (*n* = 38; 59.4%) or remained stable (*n* = 35; 57.4%) were those who were classified as normal. During the pandemic, most of the people who gained weight (*n* = 113; 58.2%) while WFH had an increased nHDI-14 score, while those who lost weight (*n* = 53; 82.8%) and remained stable (*n* = 33; 54.1%) had a decreased nHDI-14 score (*p* < 0.001). Most of the people who gained weight (*n* = 162; 83.5%) increased their daily sedentary time, and most of them (*n* = 126; 64.9%) were people who did not engage in physical activity (*p* < 0.001). On the other hand, most of those who lost weight (*n* = 42; 65.6%) had also increased their daily sedentary times, while the majority of them (*n* = 39; 60.9%) were those who performed physical activity (*p* < 0.001).

According to the multinomial logistic regression model (Table 4), participants whose work shifts remained stable (OR [95% CI]: 2.398 [1.017–5.657], *p* = 0.046) were more likely to gain weight compared to those who had increased work shifts during WFH. People with underweight and normal BMI classifications had a protective factor in gaining weight compared to those who were obese, and they reduced the risk (1 − 0.009) = 99% in those who were underweight and (1 − 0.115) = 88% in those who were normal (underweight BMI OR [95% CI]: 0.009 [0.001–0.084], *p* < 0.001; normal BMI OR [95% CI]: 0.115 [0.026–0.517], *p* = 0.005). Similarly, when evaluated in terms of physical activity, a statistically significant difference was observed in those who gained weight compared to those who lost weight (not physically active OR [95% CI]: 2.276 [1.124–4.609], *p* = 0.022). Lack of physical activity during the pandemic was a risk factor for weight gain compared to those who lost weight.

On the other hand, there was a statistically significant difference between those whose weight remained stable and those who lost weight, whose marital status was single, and whose daily sedentary time was reduced (marital status OR [95% CI]: 0.237 [0.079–0.709], *p* = 0.010; changes in daily sedentary time OR [95% CI]: 0.117 [0.024–0.583], *p* = 0.009). No statistically significant difference was found in terms of other variables in the model.

When the physical activity patterns of the participants were evaluated, individuals who did not routinely perform physical activity dropped to 188 from 202 during the COVID-19 pandemic. Individuals who engaged in physical activity (*n* = 140; 42.7%) and did not (*n* = 188; 57.3%) during the COVID-19 pandemic were compared (Table 5). BMI was calculated as 25.35 kg/m^2^ in those who did not engage in physical activity and 23.30 kg/m^2^ in those who did (*p* = 0.009). Individuals who engaged in physical activity during the COVID-19 pandemic had a 5.75 points higher pHDI-10 score (*p* < 0.001) and 3.04 points lower nHDI-14 score (*p* = 0.006). No significant differences were found in the average daily sedentary time (*p* = 0.435) and PSS score (*p* = 0.469) between the physical activity and no physical activity groups.

## 4. Discussion

This study aimed to evaluate the relationship between WFH and possible changes such as dietary behavior, sedentary lifestyle, physical activity, and stress during the COVID-19 pandemic. It is revealed that WFH led to increased sedentary lifestyle, stress, and snacking throughout the day, resulting in gaining weight.

A new study on employees and employers who switched to WFH during the pandemic reported that at least 16% of this group will continue, even if the COVID-19 pandemic ends [26]. According to the results of the same study, although the number of WFH employees is high, there are significant differences between sectors and employees with higher education levels and better incomes. These results are in line with our current study in which the majority of the participants were teachers; academics; engineers; or banking, finance, or accounting professionals. In a study that analyzed emails and meetings of 3.1 million people WFH, including 16 countries around the world, it was found that the average workday increased by 8.2% (48.5 min) in the first weeks of the pandemic [27]. Similarly, 46% of the individuals participating in our study reported that their daily working time increased.

The results of this study are consistent with other studies that demonstrated a positive relationship between weight gain and quarantine measures [10,28]. Most people who gained weight while WFH were overweight and obese, had an increase in nHDI-14 score, did not engage in physical activity, and had an increase in daily sedentary time. Similarly, studies conducted on weight gain are associated with unhealthy dietary changes, decreased physical activity, and increased alcohol consumption [29,30]. The body weight of 59.1% of individuals increased while WFH, and this increase was higher in individuals who were overweight and obese in the current study. In a study conducted during the first 10 weeks of lockdown in Turkey, 35% of participants gained weight [31]. Additionally, a different study including the first 40 weeks of lockdown in Turkey showed that 58.5% of the participant gained weight [32]. It was determined that individuals who were already overweight and obese at the beginning of the pandemic gained more weight during the lockdown periods [33,34,35,36,37]. Considering that Turkey has a high obesity incidence of 32.1% [38,39], weight gain may be a more likely risk for the Turkish population who practice WFH than in European countries.

Studies conducted on eating behaviors during the COVID-19 lockdown revealed that the consumption of canned and packaged foods such as chocolate, confectionery, biscuits, and cakes increased, and the consumption of vegetables and fruits decreased [10,40]. On the other hand, some positive changes were detected in some studies, such as decreased consumption of processed meat and carbonated beverages, as well as increased consumption of fish, white meat, fresh vegetables, and fruits [12,40]. Similarly, the results obtained from this study support previous findings that both positive and negative changes were observed in eating habits. Follow-up studies are needed to better understand the mechanisms underlying healthy or unhealthy dietary behavior changes.

There was no significant difference between before and during scores of nHDI-14 and pHDI-10. At the same time, 42.7% of the participants reported that their diet did not change. Previous research has indicated that daily habits affect eating behaviors [41,42]. In this study, despite major changes in daily life such as the change in work life and the stress factors caused by the COVID-19 pandemic, no change was found in the diet quality and dietary behavior of individuals before and during COVID-19. According to the results of a study examining individual, household, and sociodemographic factors that cause changes in food consumption and diet quality in Brazilian adults during the COVID-19 pandemic, despite the rapid and unexpected changes in daily life with the COVID-19 pandemic, most of the participants maintained their pre-specified diet (healthy or unhealthy) [43].

The barriers to physical activity (such as the closed gyms) and the transition to WFH have led to an increase in the average daily sedentary time and a decrease in physical activity [44]. Most of the studies examined in a systematic review revealed that physical activity levels decreased significantly with the increase in daily sedentary time during the COVID-19 quarantine, and the transition to WFH arrangements was highlighted as the reason for this situation [45]. The results of this study show that a lack of physical activity is an important risk factor for weight gain. In addition, other studies have shown that this situation is a major risk factor for weight gain [18,32,40,46,47].

In a study examining the relationship between sedentary behaviors and the incidence of unhealthy diet during the COVID-19 quarantine in Brazil, it was emphasized that people with a higher frequency of watching TV were more likely to have unhealthy eating behaviors [48]. The current findings indicate that individuals who engaged in physical activity were found to have a lower BMI and Non-Healthy Diet Index score and a higher Pro-Healthy Diet Index score compared to those who were not physically active. Individuals who exercise also care about healthy eating. It was determined that those who did not engage in physical activity had a significantly higher BMI.

In our study, the PSS score of the participants increased by 4.7 points after WFH. In addition, individuals who gained weight were found to have higher PSS scores. These findings are in line with previous research reporting that consuming unhealthy foods is significantly associated with stress and depressive symptoms [49,50]. In this context, it is thought that the stress related to the COVID-19 pandemic may lead to an increase in unhealthy eating habits and may lead to the development of a more negative mood in individuals during the pandemic. In the study of Tribst et al. (2021), it was reported that the diet quality of people with positive emotions such as peace, faith, and confidence improved, while the diet quality of people with negative emotions such as anxiety, fear, mental fatigue, and stress worsened during the COVID-19 pandemic [43].

Our study has some limitations. Conclusions drawn were based on self-reported amounts of food frequencies. The study was designed as cross-sectional; however, some questions were related to the pre-pandemic period. Responses to these questions may potentially have errors due to imperfect recalling. Additionally, there are limitations regarding the recall of dietary behavior before the COVID-19 pandemic. In fact, while web-based nutrition cohort studies during the pandemic are recognized as a rich resource for generating new and timely evidence, the analyses included in the study need to be interpreted with caution [50]. There are also several strengths in this study. To our knowledge, there are no studies examining the effects of WFH on lifestyle habits with respect to eating behavior changes resulting in weight gain. In addition, we collected data on dietary behavior, food frequencies, stress, physical activity, and sedentary lifestyle that is important in further understanding how WFH affects employers’ lifestyles. Further, lifestyle factors that are the consequences of COVID-19, such as unhealthy dietary behaviors, increased sedentary time, and stress status, were interrelated and interconnected and therefore should be evaluated together, as we did in this study.

## 5. Conclusions

To the best of our knowledge, this is the first study about WFH and related detailed lifestyle factors and it will set a baseline for future studies. The data obtained from this study not only provide information about nutrition, physical activity, and stress status of people WFH but also will guide them while advising on adapting to a new lifestyle. The results of this study draw attention to the fact that WFH leads to increased sedentary lifestyle, stress, and snacking throughout the day, resulting in gaining weight. Finally, further research and initiatives are needed on how to manage and support the different lifestyle factors affected by WFH, which is thought to be on the agenda after the pandemic.

## Figures and Tables

**Figure 1 nutrients-14-04006-f001:**
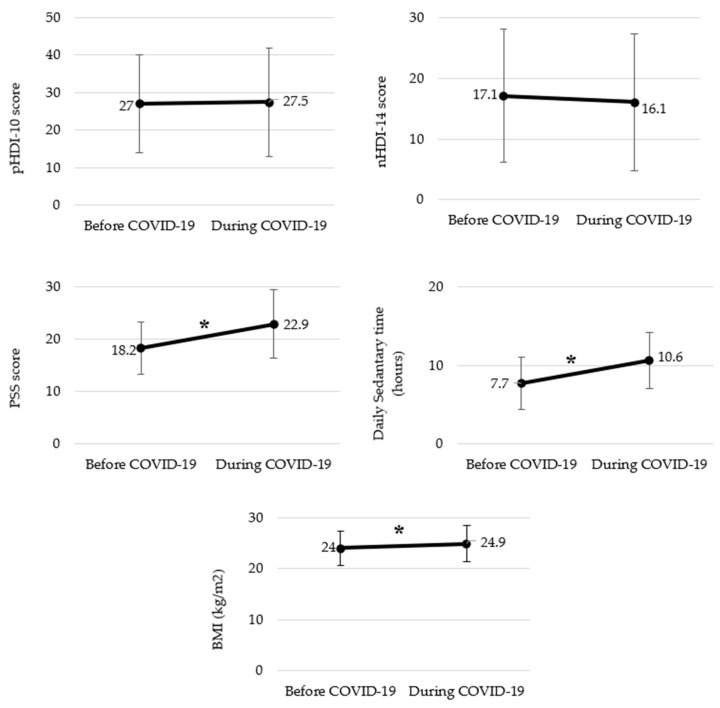
Changes due to COVID-19 in BMI; pHDI-10, nHDI-14, and PSS scores; and daily sedentary time among individuals who work from home. ** p* < 0.05 obtained from Wilcoxon signed rank test. Descriptive statistics are expressed as mean. Abbreviations: COVID-19, coronavirus disease 2019; BMI, body mass index; pHDI-10, Pro-Healthy Diet Index (pHDI-10); nHDI-14, Non-Healthy Diet Index; PSS, Perceived Stress Scale.

**Table 1 nutrients-14-04006-t001:** Socio-demographic characteristics of the study participants (*n* = 328).

Characteristics	Frequency (*n*)	Percentage (%)
Gender		
Female	260	79.3
Male	68	20.7
Age (mean ± SD) (year)	31.3 ± 8.3	
Marital status		
Married	141	43.0
Single	187	57.0
Child under care		
Yes	97	29.6
No	231	70.4
Occupational group		
Teaching	107	32.6
Architecture and engineering	58	17.7
Business and financial operations	55	16.8
Arts, design, and media	23	7.0
Management	11	3.4
Psychologist	10	3.0
Others	64	19.5
Work shift		
Stable	106	32.3
Increased	151	46.0
Decreased	71	21.6
Weight (kg)	69.7 ± 15.1	
Height (m)	167.1 ± 7.8	
BMI (mean ± SD) (kg/m^2^)	24.9 ± 4.6	
Underweight	18	5.5
Normal	157	47.9
Overweight	109	33.2
Obese	44	13,4
Weight status		
Stable	61	18.6
Lost weight	64	19.5
Gained weight	194	59.1
Do not know	9	2.7

Abbreviations: SD, standard deviation; BMI, body mass index. Descriptive statistics are expressed as frequency, percentage, or mean (standard deviation).

**Table 2 nutrients-14-04006-t002:** Comparison of mean weight changes; BMI; pHDI-10, nHDI-14, and PSS scores; and daily sedentary time of individuals who lost or gained weight during the COVID-19 pandemic.

	Weight Gain (*n* = 194)	Weight Loss (*n* = 64)	*p*-Value
During the COVID-19 Pandemic			
Average weight change (kg)	5 (2–30)	5 (1–45)	0.927
BMI (kg/m^2^)	25.70 (17.6–40.5)	22.80 (16.6–31.6)	*p* < 0.001 *
pHDI-10 score	24.75 (1.80–100)	22.30 (0–66.70)	0.452
nHDI-14 score	17.07 (0.43–85.71)	7.89 (0–35.14)	*p* < 0.001 *
PSS score	23 (7–40)	23 (1–38)	0.416
Daily sedentary time (hours)	10 (2–24)	9 (2–20)	0.028 *

Abbreviations: COVID-19, coronavirus disease 2019; BMI, body mass index; pHDI-10, Pro-Healthy Diet Index (pHDI-10); nHDI-14, Non-Healthy Diet Index; PSS, Perceived Stress Scale. * *p* < 0.05 obtained from Mann–Whitney U test. Descriptive statistics are expressed as median (min–max).

**Table 3 nutrients-14-04006-t003:** Bivariate analyses of weight changes after WFH and potential predictor variables.

During the COVID-19 Pandemic	Weight Loss (*n* = 64)	Weight Gain (*n* = 194)	Weight Stable (*n* = 61)	*p*-Value
	*n* (%)	*n* (%)	*n* (%)	
Gender				
Female	52 (81.3)	148 (76.3)	51 (83.6)	0.407
Male	12 (18.8)	46 (23.7)	10 (19.1)	
Age group				
19–39	53 (82.8)	164 (84.5)	50 (82)	0.873
40–63	11 (17.2)	30 (15.5)	11 (18)	
Marital status				
Single	41 (64.1)	115 (59.3)	29 (47.5)	0.147
Married	23 (35.9)	79 (40.7)	32 (52.5)	
Child				
No	46 (71.9)	135 (69.6)	46 (75.4)	0.675
Yes	18 (28.1)	59 (30.4)	15 (24.6)	
Work shift				
Stable	15 (23.5)	73 (37.6)	18 (29.5)	0.259
Decreased	17 (26.6)	36(18.6)	14 (23)	
Increased	32 (50)	85 (43.8)	29 (47.5)	
BMI				
Underweight	6 (9.4)	3 (1.5)	9 (14.8)	*p* < 0.001 *
Normal	38 (59.4)	79 (40.7)	35 (57.4)	
Overweight	17 (26.6)	78 (40.2)	12 (19.7)	
Obese	3 (4.7)	34 (17.5)	5 (8.2)	
Changes in pHDI-10 score				
Decreased	22 (34.4)	83 (42.8)	32 (52.5)	0.338
Increased	37 (57.8)	100 (51.5)	25 (41)	
Stable	5 (7.8)	11 (5.7)	4 (6.6)	
Changes in nHDI-14 score				
Decreased	53 (82.8)	79 (40.7)	33 (54.1)	*p* < 0.001 *
Increased	10 (15.6)	113 (58.2)	27 (44.3)	
Stable	1 (1.6)	2 (1)	1 (1.6)	
Changes in PSS score				
Decreased	14 (21.9)	29 (14.9)	7 (11.5)	0.461
Increased	44 (68.8)	142 (73.2)	49 (80.3)	
Stable	6 (9.4)	23 (11.9)	5 (8.2)	
Changes in daily sedentary time				
Decreased	16 (25)	14 (7.2)	4 (11.8)	*p* < 0.001 *
Increased	42 (65.6)	162 (83.5)	42 (68.9)	
Stable	6 (9.4)	18 (9.3)	15 (24.6)	
Physical activity				
No-physical activity	25 (39.1)	126 (64.9)	31 (50.8)	*p* < 0.001 *
Physically active	39 (60.9)	68 (35.1)	30 (49.2)	

Abbreviations: COVID-19, coronavirus disease 2019; BMI, body mass index; pHDI-10, Pro-Healthy Diet Index (pHDI-10); nHDI-14, Non-Healthy Diet Index; PSS, Perceived Stress Scale. * *p* < 0.05 obtained from Fisher–Freeman–Halton test. Descriptive statistics are expressed as frequency (percentage).

**Table 4 nutrients-14-04006-t004:** Multinomial logistic regression model of changes in body weight and potential predictor variables.

Weight Gain during WFH					
	β	OR	*p*-Value	Lower 95% CI	Upper 95% CI
Sex (ref.: male)					
Female	−0.124	0.883	0.803	0.332	2.349
Age (ref.: 40–63)					
19–39	0.494	1.639	0.792	0.042	64.047
Marital status (ref.: married)					
Single	−0.465	0.628	0.376	0.224	1.758
Child (ref.: yes)					
No	0.408	1.504	0.542	0.406	5.569
Work shift (ref.: increased)					
Stable	0.875	2.398	0.046 *	1.017	5.657
Decreased	0.026	1.026	0.955	0.420	2.508
BMI (ref.: BMI obese)					
Underweight	−4.676	0.009	<0.001 *	0.001	0.084
Normal	−2.162	0.115	0.005 *	0.026	0.517
Overweight	−1.431	0.239	0.062	0.053	1.073
Changes in pHDI-10 score (ref.: stable)					
Decreased	0.870	2.387	0.226	0.584	9.758
Increased	0.305	1.357	0.661	0.346	5.311
Changes in nHDI-14 score (ref.: stable)					
Decreased	0.511	1.667	0.722	0.100	27.673
Increased	2.658	14.274	0.070	0.804	253.436
Changes in PSS score (ref.: stable)					
Decreased	−0.140	0.869	0.841	0.222	3.405
Increased	−0.230	0.795	0.694	0.253	2.493
Changes in daily sedentary time (ref.: stable)					
Decreased	−0.846	0.429	0.246	0.103	1.791
Increased	0.655	1.925	0.274	0.595	6.231
Physical activity (ref.: physical activity)					
Not physically active	0.822	2.276	**0.022 ***	1.124	4.609
**Weight Stable during WFH**					
	**β**	**OR**	***p*-Value**	**Lower 95% CI**	**Upper 95% CI**
Sex (ref.: male)					
Female	0.266	1.305	0.669	0.385	4.425
Age (ref.: 40–63)					
19–39	0.258	1.295	0.900	0.023	71.750
Marital status (ref.: married)					
Single	−1.440	0.237	0.010 *	0.079	0.709
Child (ref.: yes)					
No	1.243	3.466	0.100	0.787	15.271
Work shift (ref.: increased)					
Stable	0.415	1.515	0.414	0.559	4.104
Decreased	0.253	1.287	0.620	0.474	3.499
BMI (ref.: BMI obese)					
Underweight	−0.957	0.384	0.372	0.047	3.141
Normal	−0.915	0.401	0.303	0.070	2.285
Overweight	−1.517	0.219	0.097	0.037	1.317
Changes in pHDI-10 score (ref.: stable)					
Decreased	0.647	1.910	0.430	0.383	9.538
Increased	−0.059	0.943	0.942	0.196	4.529
Changes in nHDI-14 score (ref.: stable)					
Decreased	−0.265	0.767	0.866	0.035	16.707
Increased	1.175	3.239	0.463	0.140	74.810
Changes in PSS score (ref.: stable)					
Decreased	0.154	1.166	0.858	0.216	6.312
Increased	0.843	2.324	0.256	0.542	9.969
Changes in daily sedentary time (ref.: stable)					
Decreased	−2.143	0.117	0.009 *	0.024	0.583
Increased	−1.102	0.332	0.072	0.100	1.103
Physical activity (ref.: physical activity)					
Not physically active	0.292	1.339	0.479	0.597	3.006

Abbreviations: COVID-19, coronavirus disease 2019; BMI, body mass index; pHDI-10, Pro-Healthy Diet Index (pHDI-10); nHDI-14, Non-Healthy Diet Index; PSS, Perceived Stress Scale; CI, confidence interval; OR, odds ratio; ref, reference. * Overall significance of model *p* < 0.001, Nagelkerke R2 = 0.590, AIC (Akaike information criteria) = 487.14, dependent variable reference category: weight loss.

**Table 5 nutrients-14-04006-t005:** Comparison of BMI; pHDI-10, nHDI-14, and PSS scores; and daily sedentary time depending on physical activity status during the COVID-19 pandemic.

	No Physical Activity (*n* = 188)	Physically Active (*n* = 140)	*p*-Value
During COVID-19 Pandemic			
BMI (kg/m^2^)	25.35 (16.6–40.5)	23.30 (16.4–46.9)	0.009 *
pHDI-10 score	23.10 (1.80–75)	28.85 (0–100)	*p* < 0.001 *
nHDI-14 score	15.21 (1.93–75)	12.17 (0–85.71)	0.006 *
PSS score	23 (1–40)	22.5 (7–40)	0.469
Daily sedentary time (hours)	10 (2–24)	10 (2–22)	0.435

Abbreviations: COVID-19, coronavirus disease 2019; BMI, body mass index; pHDI-10, Pro-Healthy Diet Index (pHDI-10); nHDI-14, Non-Healthy Diet Index; PSS, Perceived Stress Scale. * *p* < 0.05 obtained from Mann–Whitney U test. Descriptive statistics are expressed as median (min–max).

## Data Availability

Not applicable.

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
