# Peer review of "Working from Home during the COVID-19 Pandemic and Its Effects on Diet, Sedentary Lifestyle, and Stress"

_nutrients, 2022, doi:10.3390/nu14194006_

Round 1
Reviewer 1 Report
This is an interesting research paper that addresses an important public health issue: “Working from Home during COVID-19 Pandemic and Its’ Effects on Diet, Sedentary Lifestyle, and Stress
Overall, I feel that the topic of this paper is important and timely. However, there are some minor concerns with the paper as outlined below.
Introduction
Throughout the introduction, there is a lack of bibliographic references i.e. line 52 “Other occupational health problems associated with the intensity of WFH have been reported as
musculoskeletal pain, problems arising from isolation and a closed environment, depression, work fatigue, and burnout.” References
Material and Methods
Line85 ....” Therefore, we aimed to explore what is the relationship between working from home...” the authors have previously used acronyms to designate working for home (WFH) , they should not write in full again.
Line 98 ...” Following the construction process of the questionnaire form, it was uploaded to Google Forms and delivered to the volunteered participants to whom the researchers reached through individual relationships via social media channels, emails, and messaging apps with a
special target of employees whose work from home”.
The authors do not describe the use of any procedure to control the number of times each participant could respond. This may be a study bias, so I suggest that if you used a procedure, mention it, otherwise you should present it as a study limitation in the discussion.
Results
The results are well presented
Discussion
In the discussion, it is unclear what the main findings of the study really were.
Line 377- to improve the de paper, I suggest in the limitations of the study, the authors should refer to the recall bias for data reporting. The study is cross-sectional (data collection at one point in time, but it presents questions related to two different times (before and during COVID)
Author Response
Dear Reviewer 1,
Thank you for your consideration and comments on our manuscript entitled “Working From Home during COVID-19 Pandemic and Its’ Effects on Diet, Sedentary Lifestyle, and Stress”.
The introduction was updated to provide sufficient background and include all relevant references. Regarding your comment on our introduction; there were missing references due to the same reference for consequent sentences. We added missing references to those sentences.
To explain the control of only one response for each participant; Google Form has an option that says ”Allow only one response per user.” This option is enabled for a Google Form, respondents will have to sign in with their Google account to access the form. Their account information wouldn’t be seen by Google Form creators/researchers. This explanation was also added to the methodology.
Overall main findings were summarized in the first paragraph of the discussion to make it clearer. Recall bias and further study design limitations were also mentioned.
Thank you very much for your time.
Sincerely,
Merve Guney-Coskun

Reviewer 2 Report
I read the manuscript entitled: "Working from home during COVID-19 pandemic and its effects on diet, sedentary lifestyle, and stress". Unfortunately, the article lacks originality. During the previous two years, many similar articles on the same topic were published. In addition, the authors failed to determine the difference between their article and the previously published articles.
Author Response
Dear Reviewer 2,
Thank you for your consideration and comments on our manuscript entitled “Working From Home during COVID-19 Pandemic and Its’ Effects on Diet, Sedentary Lifestyle, and Stress”.
Regarding your comment on our manuscript; Although there are many studies investigating the behavioral changes prominent in this period in the general population, it is known that additional studies are needed on individuals in specific groups and conditions. COVID-19 seems to be far behind since the first measures which included working from home. Many employees started to WFH on COVID-19 and many of them continue independent of COVID-19. WFH is unlikely to be a temporary phenomenon, there is a need to determine changes in lifestyle for further nutritional advice for better lifestyle habits.
To our knowledge, there are no studies examining the effects of WFH on lifestyle habits with respect to eating behavior changes resulting in weight gain. In addition, we collected data on dietary behavior, food frequencies, stress, physical activity, and sedentary lifestyle which is important in further understanding how WFH affects employers' lifestyles. Those arguments were also mentioned in our manuscript.
Thank you very much for your time.
Sincerely,
Merve Guney-Coskun

Reviewer 3 Report
Dear author this is an interesting study and a well-written manuscript. I would suggest a thorough English editing for minor issues.
Furthermore, on L. 322-323 the sentence is not clear. In my understanding, the meaning is that 35% and 58,5% of the participants experienced weight gai within a period of 10 and 38-40 weeks, respectively. Please consider rephrasing.
Finally, there are more relevant studies, e.g. in Cyprus about the impact of COVID-19 preventive measures on the persons’ lifestyle, which are not included in the references (Athanassiou; Solomou; etc). Please consider adding them.
Author Response
Dear Reviewer 3,
Thank you for your consideration and comments on our manuscript entitled “Working From Home during COVID-19 Pandemic and Its’ Effects on Diet, Sedentary Lifestyle, and Stress”.
Regarding your comment on our manuscript; The introduction was updated to provide sufficient background and include all relevant references. English spelling and grammar were checked once again. Rephrasing was made.
An article that provides information regarding anxiety and depression symptoms during COVID- 19 from Solomou et al. was added to the reference list by your suggestion.
Thank you very much for your time.
Sincerely,
Merve Guney-Coskun

Round 2
Reviewer 2 Report
The authors improved the quality of the manuscript